# Lecturers' information literacy experience in remote teaching during the COVID-19 pandemic

**Heriyanto** *[☯], **Lydia Christiani**[‡], **Rukiyah**[‡]

Library Science, Faculty of Humanities, Diponegoro University, Semarang, Central Java, Indonesia

☯ These authors contributed equally to this work.
‡ LC and R also contributed equally to this work.
* heriyanto@live.undip.ac.id

**Data Availability Statement:** All relevant data are within the manuscript and its Supporting Information files.

**Funding:** The study funded by Universitas Diponegoro for Heriyanto, Rukiyah, and Lydia

## Abstract

The advent of coronavirus disease 2019, or COVID-19, continues to trigger several important disruptions/innovations in practically every sector around the world. Additionally, the impacts are predominant in certain educational systems and in creating opportunities. Previous studies had addressed possible effective methods in handling distant learning and student interactions. This qualitative study explored lecturers' information literacy experience during online classes as a result of the pandemic. Semi-structured interview techniques were applied among participants, made up of 15 lecturers in the Humanities Faculty, Diponegoro University, Indonesia, and thematic analysis was used to analyze the data obtained. The results showed the focus of lecturers' information literacy experience was primarily on student interactions and knowledge of various online learning platforms. However, information repackaging was a significant initial consideration during virtual classes, after identifying salient student characteristics. In summary, the present study have contributed to the theoretical understanding of information literacy and may be of benefit to the teaching faculties for enhancing teaching and learning activities, as well as providing student support.

## Introduction

The impact of COVID-19 on universities has been widespread [1, 2]. In June 2020, the Indonesian government reported the transition of major higher education institutions to various alternative teaching and learning modes, over the past three months. Essentially, all universities considered implementing online classes as a means to curtail the virus spread effectively [3]. However, the majority of lecturers perceive this approach as a novel initiative, but with very limited instructions in place. As a consequence, the online tuition requirement created a unique predicament for the country's higher education systems. Moreover, without adequate preparation, lecturers possibly become prone to facing numerous complex issues [4, 5]. Particular previous studies have demonstrated that this virtual concept demands a separate pedagogy from a traditional classroom approach [6, 7].

Online teaching amid the growing COVID-19 pandemic and the influence on teaching and learning techniques, specifically in higher education, have been investigated globally [1, 8–10].

Christiani. The grant number is 1215-26/UN7.5.6/HK /2020. However the funders had no role in study design, data collection and analysis, decision to publish, or preparation of the manuscript.

**Competing interests:** The authors have declared that no competing interests exist.

These studies reported the unique diversity of emergency remote teaching compared to conventional online mode [8] and the way faculties engage with related tools to promote student interactions while revealing faculties' perspectives about remote teaching [11, 12]. Presently, no report exists on lecturers' information literacy experience during online tuition. Therefore, the purpose of this paper is to investigate and report lecturers' information literacy experience during emergency remote teaching in the Faculty of Humanities, Diponegoro University. Furthermore, academic activities have migrated to virtual mode across the country as of March 2020. This movement was an urgent situation, where the lecturers were allocated little time to prepare. Under this condition, the study's objective tends to vary from online programs under normal circumstances.

## Literature review

### Emergency remote teaching

The current global circumstance in terms of the transition from conventional teaching and learning systems in higher education to online mode, as an emergency delivery mode, has been identified by certain experts. Iglesias-Pradas *et al.* [13] emphasized "the difference between online teaching and emergency remote teaching lies in that online results from careful instructional design and planning, requiring an investment in a whole ecosystem of learner support that takes time to build" [para. 6]. Meanwhile, the emergency remote option presents as a form of temporary shift of instructional delivery to an alternative channel in response to crisis situations [8]. The word 'temporary' is highlighted as the teaching mode assumed is delivered onsite after the setback.

No single faculty around the world expected this urgent shift [13]. People with previous virtual experience showed more adaptability, although several lecturers indicated no prior online knowledge, and possibly encountered certain challenges, including insufficient preparation. Consequently, new remote teaching methods have had to be developed, with or without experience, as these means require a different pedagogy from onsite classroom arrangement [1, 12]. In addition, considerations are also needed to initiate relevant adjustments to assignments and exams [11].

Furthermore, other studies focused on the employment of online teaching tools and instructional modes. In relation to synchronous and asynchronous virtual deliveries, Iglesias-Pradas *et al.* [13] reported a greater preference for resource materials in a synchronous pattern, as faculties and students were physically distant, but communicating in real time using a range of videoconferencing applications, similar to certain traditional classes. Academic lectures tend to choose familiar methods and tools. Two main online resources commonly used by faculty include a learning management system (LMS) and a video conferencing platform [14]. Lecturers use LMS to distribute course content and related academic materials. Also, it is possible to integrate discussions during and after lectures, in addition to conducting exams and post-assignment details [15]. The major limitation referred to the short time allocated to learning the platform, therefore, recommendations were suggested to provide continuous training on the use of educational tools toward enhancing faculties' capacity [13, 16].

Based on the prevalent issues and challenges of emergency remote teaching, concerns have been generated regarding the tuition quality in higher education. Online training is not expected to inspire confidence with a lecturer talking to a screen. This only creates a teacher centered learning; instead, there is need to provide timely feedback to students' questions and comments [10]. In participating in remote course delivery, Rapanta *et al.* [10] and Rad *et al.* [1] recommended the need for active interactions between teachers and students, as everyone in the virtual classroom is capable of learning from others.

Previous studies on faculty's experience of online teaching emphasized the emerging issues related to emergency remote teaching as well as coping strategies. This current study investigated the information literacy experience of lecturers during emergency remote teaching.

## Information literacy as a learning experience

Information literacy embraces several definitions, and more importantly, the variations in understanding originated from the various theoretical perspectives [17] have influenced these research objectives [18]. Despite offering numerous descriptions, several experts relate the concept of information literacy to problem-solving activities involving critical thinking and the ability to use information in everyday life [19].

Many higher education studies regarded information literacy as an essential skill in ensuring academic success. These abilities include information sourcing, evaluation and selection, as well as ethical implementation of the knowledge amassed. Also, the skill-based concept is significantly influenced by the behavioral aspects of information literacy, considered to be a talent or aptitude an individual may possess [18].

The current study adopted the relational perspective of information literacy. This is an information literacy perspective that emphasizes the relationship between people and information when learning in a different context [20]. From this perspective, information literacy focuses on the different ways people relate to information and interact with the information world [21]. Bruce's work [1997] is significant for this study because it moved the understanding of information literacy beyond skills to show the complexity and nuanced elements of the phenomenon. Information literacy from the relational perspective is seen to be more than just developing skills and competencies to undertake actions or behaviors; it also involves emotion, attitudes and other more experiential elements [21–23]. As a theoretical framework for information literacy research, the relational perspective provides a wider interpretation and offers different insights into information literacy [24].

A recent study by Yates [22] employed the above approach in exploring the information literacy experience, where the variations in people's encounters on the application of information in learning about health are the topic of study. Previously, Andretta [25] utilized a similar approach to determine the experience of information literacy of undergraduate students. Sayyad Abdi, Partridge, and Bruce [26] researched web designers and developers information literacy, while Demasson [27] investigated the category of persons engaged in serious leisure activities in traditional bases.

These above-mentioned studies highlighted the information literacy experience from various backgrounds and demonstrated other means of assessment. Similarly, several opportunities were readily available in exploring information literacy in separate contexts. The present research is aimed at investigating lecturers' information literacy experience during emergency remote teaching. Therefore, this research addresses the question of how the lecturers experience emergency remote teaching as part of their information literacy.

## Method

This study employed a basic qualitative method, a method that is used to study the interaction of people in their naturally occurring conditions [28]. It involves qualitative data collection [e.g., interviews] to understand and explain social phenomena [29]. This basic qualitative method can be used by researchers who are interested in understanding the meaning that is not discovered but constructed [30] Hence, the basic qualitative method is appropriate for a study that is interested in how people interpret their experiences, how they construct their world, and what meaning they attribute to their experiences [30]. Even though this

understanding identifies all qualitative research, other types of qualitative studies have an additional dimension. For instance, a grounded theory study seeks not just to understand, but also to build a substantive theory about the phenomenon being studied [30].

This study was reviewed and validated by the Faculty of Humanities Ethics Committee, with approval number: 1215-26/UN7.5.6/HK/2020.

The research sample comprised 15 participants, lecturers of six undergraduate courses from the Faculty of Humanities, Diponegoro University, Indonesia. These programs include Library Science, Japanese Literature, Social Anthropology, Indonesian Literature, English Literature and History. Based on the study objective, a criterion was considered for acceptance, where only individuals at the lecturer level were eligible, and in this regard, assistant lecturers were not qualified. Also, the completion of six months distance training during the COVID-19 outbreak is further required as evidence of extensive knowledge for the remote arrangement.

The principal researcher also sent the recruitment flyer to a WhatsApp group of all the Faculty of Humanities' lecturers. This flyer distribution was considered the best and quickest method to disseminate necessary details. Nine lecturers agreed to participate in the interviews at the first flyer distribution. Subsequently, the principal researcher posted the flyer for the second time, and six further participants signed up. The list of participants is provided in Table 1. All participants were assigned sufficient time to ask questions. Also, an information sheet and consent form were provided to properly understand the research parameters [31]. Table 1 shows the participants of this study.

The study population was not limited to specific courses, as the overall purpose was not to simply investigate the nature of teaching from home within the information literacy experience for a specific group of teachers. Therefore, all the lecturers from six different courses from the Faculty of Humanities, Diponegoro University were targeted for data collection, using semi-structured interviews. Moreover, the principal researcher was responsible for the interviews and was the only one engaging with participants. This interviewer demonstrated adequate interrogative knowledge, as a Ph.D. graduate of Queensland University of Technology, Australia. The questions used for the interviews were:

1. Can you tell me what your subject is?

2. Can you tell me about how you deliver lectures?

3. Can you tell me how you engage with your students?

4. Can you tell me about the information sources you use for your distance teaching?

5. How would you define 'distance teaching'?

Follow-up questions were used for the interviews to explore participants' responses. One response from a participant may trigger two or three points to probe [32]. Some follow-up questions that were used during the interviews included:

1. Can you tell me more about that?

2. What did that make you think about that?

The interview guide is available in S1 File.

In accordance with established qualitative practice, the collected data were analyzed using thematic analysis. This method was applied to identify and examine patterns or themes considered significant in describing the study phenomena [14]. In addition, the approach was used to interpret various aspects of the subject matter, by "encoding" the qualitative data obtained [15]. The processes were composed of three phases, termed data familiarization, code

**Table 1. Participants of the study.**

| Participants | Courses | Sex | Notes |
|---|---|---|---|
| Rika | Course A | Female | 36-year-old lecturer with seven years' teaching experience. Rika is familiar with the basic features of the online resource platform as the main tool for engaging the students. |
| Harry | Course A | Male | 33-year-old lecturer with five years' teaching experience. Harry reported being familiar with the teaching platform as the main tool for interacting with students and has utilized virtually all the features. Harry has also used integrated YouTube videos into the platform to share with students. |
| Deny | Course A | Male | 41-year-old lecturer with five years' teaching experience. Deny, identified as a competent user of the teaching platform, is very conversant with majority of the features in interacting with students. |
| Rudi | Course B | Male | 37-year-old lecturer with nearly eight years' experience. Major features employed for teaching include text chat, audio meetings, reading files, assignments and quizzes. Rudi uses the teaching platform as a single tool in sharing resources with students and also reported being familiar with searching for information through the Internet for extra resources. |
| Dhani | Course B | Female | 33-year-old lecturer with approximately four years' experience. Dhani is a competent user of Office applications and social media platforms. She is able to use the teaching platform relatively easily since being initially introduced at the commencement of the emergency remote teaching. |
| Tina | Course B | Female | 43-year-old lecturer with seven years' experience. Tina uses blogs and Twitter effectively for searching current news related to her subjects, and also claimed the teaching platform is manageable, despite a poor internet connection. |
| Nida | Course C | Female | 35-year-old lecturer with six years' experience. Nida reported that the teaching platform was easy to learn and is very handy. However, presently Nida barely uses the basic features, including post, chat, creating new teams, audio meetings, assessments and quizzes. |
| Fikri | Course C | Male | 41-year-old lecturer with eight years' teaching experience. Fikri reported working relatively easily with the platform and occasionally prepares ready-to-use videos from YouTube as teaching content. |
| Ifah | Course D | Female | 39-year-old lecturer with seven years' experience in the course. Ifah feels very relaxed about using the teaching platform for the first time, but only on the basic features like text chat, audio and video meetings, assignments and uploading content for students. |
| Bodi | Course D | Male | 42-year-old lecturer with about eight years' experience on the course. Bodi has used different information sources like Wikipedia and any other websites during distance teaching and also shares useful information with students via the teaching platform. Bodi did not observe any technical issues, except for a poor Internet connection. |
| Fuji | Course D | Male | 37-year-old lecturer with six years' teaching experience. Fuji learned the teaching platform by himself at the start of the pandemic. Also, Fuji usually has students activate the camera during lectures to enable him to view the students' presentations. |
| Tommy | Course E | Male | 31-year-old lecturer with three years' experience teaching the course. Tommy was also an alumnus of the Humanities Faculty and feels very close to the students. This enables easier interaction via the teaching platform. He has already used sufficient teaching resources from the Internet before the pandemic and currently feels confident using the platform for teaching and sharing resource materials. |
| Ara | Course E | Male | 41-year = old lecturer with 12 years' teaching experience in this faculty. Ara presents as a competent user of Office packages and Internet browsing, particularly in looking for resource materials from YouTube, Twitter and popular search engines. He also claimed to have easily adapted to the teaching platform during the first week of online teaching. |
| Bekti | Course F | Male | 49-year-old lecturer with 14 years' teaching experience. Bekti employs several tools, including YouTube to upload custom videos and share them with students. He believes YouTube and Facebook are effective in delivering content to students and colleagues. |
| Rani | Course F | Female | 29-year-old lecturer with seven years' experience in teaching the history course. Rani is identified as a fast learner in information and communication technology tools, specifically for teaching purposes. Rani agreed to seamlessly adapt to the platform and also provide short training sessions to colleagues on ways to use the application. |

generation, and theme identification. In the first stage, interview transcripts were read two to three times to familiarize participants with the sample data and ensure proper comprehension. Subsequently, a summary was written to describe researchers' reflections after each reading. In the second phase, codes were generated to detect relevant data features. This study utilized an inductive approach to see what emerged from the data. The inductive approach to coding refers to generating code that "derive from what is in the data" [33]. Once the codes had been finalized, a codebook was developed. The codebook was an important element for this study, as it was used to organize the codes following the interpretations [34]. Table 2 shows an excerpt of coding from the codebook.

Following the coding, codes that had similar features were grouped into higher-order categories. The identified broad categories or groups were named according to their content. In this phase, the groups created and the codes allocated to each group were reviewed to determine whether the codes were located in the right group.

**Table 2. Excerpts of codebook documenting the codes.**

| Codes | Total no of references | No of interviews used in | Participant 1 | | Participant 2 | |
|---|---|---|---|---|---|---|
| | | | No of references | Line no | No of references | Line no |
| Same teaching method | 1 | 1 | 1 | 347 | | |
| Independent learning instruction | 2 | 2 | 2 | 389, 433 | | |
| Video meet preferences | 4 | 1 | 1 | 452 | | |
| Self-exploration methods are working well | 1 | 1 | 1 | 477 | | |
| Asking students to read more | 1 | 1 | | | 1 | 29 |
| Video interactions after reading | 2 | 1 | | | 1 | 115–145 |

In the third phase, themes were identified as the data patterns, believed to describe and organize the aspects of the phenomenon derived from the data. However, the previously generated codes were reviewed and examined multiple times against the main research question, phrased as, how was the lecturers' information literacy experience during remote teaching? To determine the themes, in this phase, themes were identified based on similarities and differences observed between the groups and codes. The identified themes are explained in the following section.

## Findings

The research findings show the participants' information literacy experience in emergency remote teaching during the COVID-19 pandemic. These unique experiences consist of the ways the participants familiarized themselves with the new environment of distance teaching, which also involved the use of new information technology. The participants' experience also included the various methods technology which was applied in terms of lecture delivery, research consultations, and students' research and course examinations.

### Information repackaging

This theme depicts an aspect of the participants' experience of remote teaching as part of their information literacy that was associated with recording information for their students. Lecturers' information literacy now encompasses engagement with different modes of social media platforms they used to present lectures. Remote teaching involves several interactions between lecturers and students by sharing text messages and holding video meetings for a short period on an online platform. Fuji, Rika and Nida believed these methods did not adequately convey the full intentions of the lecturers' message. However, the concerns were resolved by creating video recordings of lectures and uploading them to other platforms, e.g., Facebook or YouTube.

The files involved selecting distinct content deemed important for students in forms other than text messages. These video recordings were expected to serve as additional materials alongside e-books and journal articles to stimulate discussions during lectures. However, as the recordings were also available to the general public, other comments within the platform occured, not only between the lecturer and students.

Rudi needed to create a recording on the technical process of writing Kanji letters, and, in turn, students were required to document the practice by uploading the video to the online platform. In this manner, the lecturer can provide feedback based on individual/group performance, and also as a good knowledge-sharing exercise.

In the context of an information literacy experience, academic resources are expected to be interactive and engaging. Selecting, recording, and then uploading content to YouTube for

easy access provides a constructive opportunity for lecturer–student interactions. The purpose was not only based on getting the students' attention as their intended audience but also targeted colleagues' online views and comments via shared videos on social networks.

## Supporting students as independent learners

This theme denotes an aspect of participants' experience of remote teaching as part of their information literacy related to supporting students as independent learners. Emergency remote teaching environments have encouraged lecturers in observing minimal face-to-face contact with students, and they were are also required by the university not to spend above 30 minutes on video meetings, in consideration of internet access. Moreover, students in rural communities usually experience certain challenges including poor internet connections, inadequate modern devices, e.g., smartphones or laptops, and issues with data purchase. Therefore, 30-minute lectures were ultimately considered sufficient.

Conversely, five lecturers complained that this time was rather too short to allow for detailed interactive discussions. In consequence, Rika, Fuji, Nida, Ara and Bekti encouraged students to commit personal time to study the class materials provided on the teaching platform and observe related phenomena. Dhani regarded remote teaching as a hindrance to students' overall learning objectives, but, on the other hand, possibly attracts new opportunities for the lecturers to encourage independent studying.

*They don't have to rely mainly on the lecturer, I think, and I don't have to teach them every single step to understand the Japanese text. I really hope they would be able to explore, and be creative in reading and to present what they have found from the literature available on the Internet to the class in the next online meeting* (Dhani)

Furthermore, about the course materials, the lecturers discovered the need to encourage self-study by offering valuable resources, including reading materials and videos. These provisions serve as a stimulus for guidance, and also as useful information on related supplies. Similarly, Tommy reported the additional insights generated by the students were significant for lecturers to possibly incorporate into subsequent materials. In this context, information literacy is also experienced by the lecturers as a form of providing relevant information with an intention to inspire a further search for additional related resources for self-study purposes. However, the majority of the lecturers regard the initiative as an obstacle to student interaction, by describing the use of information as activities including selecting and uploading content to the online platform, as well as monitoring student contributions.

## Familiarization with the teaching platforms

This theme illustrates an aspect of participants' experience of remote teaching as part of their information literacy related to their effort of making them familiar with the teaching platforms. The courses offered by the Faculty of Humanities include culture, language and literature studies. The subjects require active lecturer-student interactions, as there is a need to deliver and assess particular skills, e.g., speaking, or listening. The ability to achieve the learning goals in these topics relies heavily on interactions between lecturer and student as well as among students. Bodi noted the selection of a teaching platform is essential for an effective learning process, based on individual course needs. The current online platform has been identified by certain participants as adequate to cater for the educational requirements and also allows for audiovisual content as well as text messages. Furthermore, Bodi stated that the

future development of any remote channel by universities is expected to consider lecturers' needs, in terms of teaching and learning objectives.

Ara reported an elaborate delivery on the online platform, using various instructional methodologies and content. Previously, YouTube content was rarely consumed, but currently, Ara is expected to obtain a range of digital resources to enable a smooth course delivery. Also, YouTube has been one of the main selections, compared to other providers of educational resources. Therefore, the teaching platform presently employed by the university was considered very helpful, probably due to easy integration with other platforms, including YouTube, Forms, and any other learning tools known to support effective teaching activities.

Conversely, based on Bekti's experience in using the platform, Bekti and her colleagues encountered initial difficulties understanding the platform. Moreover, "online teaching", appeared as a common concept, but none of the participants had ever employed the application. As a consequence, the efforts required to familiarize themselves with the provision varied in terms of timeframes and techniques. For instance, Nida, Ifah, Dhani, Bekti and Harry commented on being new users to the platform, but they did not take long to become conversant with the features. These individuals formed a study group to learn how to utilize the platform effectively, as illustrated by the following quote,

> . . .everything was happening very quickly. I had not expected the need for online teaching, until the COVID-19 outbreaks. I didn't have any experience with using such a tool, or any other tools like YouTube and so on, for online teaching. I found the study group, where we, the lecturers, got together to learn how to use the platform, very helpful. (Bekti)

Rika and Bodi stated that the university had provided a tutorial video on the basics of the teaching platform and its features. This circumstance, however, did not enable detailed comprehension.

> It doesn't talk much about the [platform] features. I think it was intended only to show how to locate it in the system, log in, and introduce the main functions like creating classes, adding students in, and uploading files. Yeah it's not that detailed (Rika).

Familiarization with the online platform was the way lecturers experienced information literacy. The lecturers regard information as their ability to essentially use the teaching platform, study groups and video tutorial offered by the university. Also, the participants described the use of the information as an initiative to join the study group to extensively comprehend the platform. These lecturers perceived the information-sharing communities as a means of fully understanding the platform and being successful in managing the subjects during classes.

## Active learners

Active learners denote an aspect of the participants' experience of remote teaching as part of their information literacy related to supporting students to become active learners. The lecturers identified that students were more active when they were invited for text chat discussions, compared to classroom settings, as several questions were generated, with corresponding responses. Also, Fuji and Bodi identified this behavior as related to the local culture, where students from Javanese cultural backgrounds were taught to always respect and obey teachers and, as a result, were expected to listen rather than ask questions during video sessions. However, as the lecturers commenced interaction through text chat medium, students were further encouraged to express any question by commenting about the delivered topics, and through this approach, great responses were recorded.

*. . .during the video calls, I always encourage them to raise any questions. But if students think that it is not polite to interrupt me, I ask them to write down their questions or comments via the text chat. I did notice, though, that a few students were brave enough to ask their questions during the video meeting, but, mostly they only did that through the text chat because they didn't want to interrupt me* (Fuji).

Initiating text discussions illustrated the lecturers' information literacy experience, where information was regarded as students' culture and attitude. The use of information was described as inviting students to participate in text discussions after video meetings through the online platform. Lecturers initiated text discussions to give an opportunity to students who interact less during video meetings. Hence, it was an effective strategy because more students were willing to participate in the discussion.

## Discussion

The results of this study showed each lecturer with a unique information literacy experience involving the new remote teaching platform. The majority of the lecturers confirmed that the idea of a fully remote resource was a brand-new academic experience. Therefore, there is a crucial need to adapt to this new technology. Also, the process of familiarizing was relatively convenient for certain lecturers, while for others, particularly the senior lecturers, the process was extremely stressful, as many were not prepared. This finding contributed to Mohamad, Salleh, and Salam's [16] studies, where the use of information technology to enhance teaching and learning activities was very limited.

There were different methods in becoming familiar with the new shift. In this study, a majority of the lecturers voluntarily joined a study group formed by colleagues, to master the platform. This report supported Somerville and Bruce's [35] study, where learning, a product of a collaborative, peer-oriented environment, was regarded as essential in certain cultures. Based on Lloyd [36], it became evident that people in local communities create an information landscape within the environment and develop social connections at work. The lecturers have begun to comprehend the emergence of vital information from anywhere or any individual. This finding matched the report by Sayyad Abdi *et al.* [26], as information literacy was also experienced by participating in similar learning communities, where information sharing breeds valuable engagement and mutual studying.

Pertaining to information repackaging into video format, the results of this research showed that faculties not only attempted to devise ways to deliver lectures effectively, but also endeavored to customize the content to match students' characteristics. Current learners were identified as "digital natives", as denoted by Prensky [37], which referred to persons born in or after 1980. In addition to this category and due to wide exposure to technology, new learning styles were easily developed, with knowledge in multiple media types and by appreciating the different forms of ICT currently utilized for learning. Under these circumstances, several participants opted to record the lectures and upload to YouTube, while others searched for relevant videos as teaching materials. The subsequent finding confirmed the results of Johnston, Barton, Williams-Pritchard, and Todorovic [38], where the majority of the undergraduates regained confidence and properly engaged YouTube videos, due to the extent of accessible and interactive information in small measures. These reports also supported Burke and Snyder's study [39], where social media tend to provide an alternative route for teachers to engage students in higher education with quality and interactive course content.

Furthermore, the results of this study contributed to Johnson, Veletsianos, and Seaman's [6] reports involving lecturers with no experience in online teaching, who were required to use

the new teaching approach. Participants observed the use of various delivery methods, including new techniques not previously utilized. Interestingly, several means to stimulate student interactions during virtual classes were devised, as simple as merely using text chat.

The lecturers demonstrated the awareness of information literacy and identified the students' preference of learning tools, including YouTube videos. These channels served as a motivation to explore other available learning materials. Also, sufficient comprehension of student interactions was developed and, more importantly, the best means of engaging these "digital native" learners. Moreover, the present study contributed to Moghavvemi, Sulaiman, Jaafar, and Kasem's [40] investigations by providing insights on the use of YouTube as a complementary tool for teaching and learning, as also reported by Johnson *et al*. [6], where lecturers were expected to employ synchronous and asynchronous video sessions.

Furthermore, by exploring a specific group of academic staff in higher education, this study is believed to enrich the investigation of information literacy by a specific group of people, in a particular workplace and profession e.g., lecturers [41, 42].

## Conclusion

This study provided new insights into lecturers' information literacy experience during the COVID-19 pandemic. The findings are believed to contribute significantly to the Faculty of Humanities, Diponegoro University and other tertiary institutions in Indonesia. These circumstances tend to enhance teaching and learning activities, as well as offer adequate student support. Also, the results may influence theoretical comprehension of information literacy, to create viable opportunities for further research in the field of emergency remote teaching and information literacy.

## Supporting information

**S1 File. The interview guide.**
(DOCX)

**S2 File.**
(DOC)

**S3 File.**
(DOCX)

**S4 File.**
(DOCX)

## Author Contributions

**Investigation:** Heriyanto.

**Methodology:** Lydia Christiani.

**Resources:** Rukiyah.

**Writing – original draft:** Heriyanto.

**Writing – review & editing:** Heriyanto.

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
