## [Decision Letter · Decision Letter 0]

17 Feb 2021

PONE-D-20-39011

Investigation into the information literacy experience of lecturers when teaching from home during the COVID-19 Outbreak

PLOS ONE

Dear Dr. Heriyanto,

Thank you for submitting your manuscript to PLOS ONE. After careful consideration, we feel that it has merit but does not fully meet PLOS ONE’s publication criteria as it currently stands. Therefore, we invite you to submit a revised version of the manuscript that addresses the points raised during the review process. I agree with the two reviewers that this study is timely and relevant. There are still some areas that need further clarification or improvement.

We look forward to receiving your revised manuscript.

Kind regards,

Mingming Zhou, Ph.D.

Academic Editor

PLOS ONE

Journal Requirements:

3. When reporting the results of qualitative research, we suggest consulting the COREQ guidelines: http://intqhc.oxfordjournals.org/content/19/6/349. In this case, please consider including more information on the number of interviewers, their training and characteristics. Moreover, please describe in more detail the methodology used for data analysis, and please provide the interview guides used as a Supplementary file.

Reviewers' comments:

Reviewer's Responses to Questions

**Comments to the Author**

1. Is the manuscript technically sound, and do the data support the conclusions?

Reviewer #1: Partly

Reviewer #2: Partly

2. Has the statistical analysis been performed appropriately and rigorously? 

Reviewer #1: N/A

Reviewer #2: N/A

3. Have the authors made all data underlying the findings in their manuscript fully available?

Reviewer #1: Yes

Reviewer #2: Yes

4. Is the manuscript presented in an intelligible fashion and written in standard English?

Reviewer #1: Yes

Reviewer #2: Yes

5. Review Comments to the Author

Reviewer #1: Your research is especially relevant and important in the current pandemic. I like that you opted to research lecturers, who often don't attain the same professional development as tenure and tenure track, professors. However, I would have liked to know more about how these lecturers were recruited; also it would have been good to have more information on each lecturer - such as years teaching, age, comfort with technology.

I also suggest a more thorough literature review. Other research exists in the area of teaching online, though perhaps without the focus on information literacy.

I also question your use of thematic analysis. From my own experience, thematic analysis does not include the use of codes. I am also not seeing any explanation of how you derived those specific codes.

Your findings are very interesting but some parts are vague. For example, "some lecturers commented that 30 minute lectures were at times to short," "most lecturers encouraged students to allocate their own timeframes," "the lecturers found that they could encourage self-study..." Can you be more specific with these findings? What do you mean by "most," or "some?" Did all lecturers find that they could encourage self-study?

Last, there are grammatical errors throughout, so this needs to be edited.

Thank you for exploring this timely issue!

Reviewer #2: Thank you for allowing me the opportunity to review this manuscript. I think given the transition of almost all higher education institutions around the world to online learning over the past year, this is a topical and relevant study. It was well written and organized logically. I do have significant concerns with the framework used to study your findings. Please see below for my suggested revisions:

1) I'm not sure you've sufficiently demonstrated why this is connected to information literacy. Although I appreciate the literature review, this doesn't highlight any formal standards or frameworks (e.g. IFLA or, in the U.S. context ACRL). It seems to me what you're describing is information behavior of lecturers? That would require a different lit review and

2) I wasn't entirely clear on the research questions - I know you were investigating how the lecturers changed their teaching based on the transition to online education, but it wasn't clear to me how this fits into the overall research framework of either information literacy or online teaching. Again, for the latter, a different or supplemental lit review and framework would be needed.

3) How do these particular lecturers represent the wider population? What were your questions? Methods need to be more specific.

Ultimately this is a good overview of a particular change in teaching due to COVID, but it wasn't especially clear to me how it fits into a larger research framework or investigation.

6. PLOS authors have the option to publish the peer review history of their article (what does this mean?). If published, this will include your full peer review and any attached files.

Reviewer #1: No

Reviewer #2: No

---

## [Author Response · Author response to Decision Letter 0]

14 Apr 2021

Editor Remarks 

1. Please ensure that your manuscript meets PLOS ONE's style requirements, including those for file naming

The manuscript has been revise following PLOS ONE’s style requirement as stated in https://journals.plos.org/plosone/s/file?id=wjVg/PLOSOne_formatting_sample_main_body.pdf and

Information about participant consent has been provided in the ethics statement in the Methods and online submission form. The written consent was informed to the participants when the participants agreed to be interviewed and before they were interviewed.

3. When reporting the results of qualitative research, we suggest consulting the COREQ guidelines: http://intqhc.oxfordjournals.org/content/19/6/349. In this case, please consider including more information on the number of interviewers, their training and characteristics. Moreover, please describe in more detail the methodology used for data analysis, and please provide the interview guides used as a Supplementary file.

• Additional information about number of interviewer, the interviewer training and characteristic have been provided following COREQ guidelines 

• The data analysis method has been described in details in the Methods section. Thematic analysis was employed in the study to interpret the qualitative data obtained from the interviews.

• The interview guide is provided as a supplementary file.

Reviewer 1 

• I would have liked to know more about how these lecturers were recruited; also it would have been good to have more information on each lecturer - such as years teaching, age, comfort with technology.

The participants recruitments technique and additional information about each lecturer have been added to the Methods section

• I also suggest a more thorough literature review. Other research exists in the area of teaching online, though perhaps without the focus on information literacy.

Thank you, this is really great suggestion. We have followed this suggestion by providing research in the area of emergency remote teaching. These researches considered relevant to this study as the context of the study is emergency remote teaching and information literacy. 

• I also question your use of thematic analysis. From my own experience, thematic analysis does not include the use of codes. I am also not seeing any explanation of how you derived those specific codes.

This study employing thematic analysis introduced by Braun & Clarke which it involves generating codes as its second phase for the data analysis. I have provided details about code generations in Method section. In addition, the details of Braun & Clarke publication on thematic analysis that employed in this study is available here Virginia Braun & Victoria Clarke (2006) Using thematic analysis in psychology, Qualitative Research in Psychology, 3:2, 77-101

• Your findings are very interesting but some parts are vague. For example, "some lecturers commented that 30-minute lectures were at times to short," "most lecturers encouraged students to allocate their own timeframes," "the lecturers found that they could encourage self-study..." Can you be more specific with these findings? What do you mean by "most," or "some?" Did all lecturers find that they could encourage self-study?

Thank you for this. The sentences have been revised with the suggested wording to be more specific with the participants.

• Last, there are grammatical errors throughout, so this needs to be edited.

Thank you for this. The revised version of the manuscript has been proofread by professional proofreading services to improve the clarity and readability. 

Reviewer 2 Remarks 

• I'm not sure you've sufficiently demonstrated why this is connected to information literacy. Although I appreciate the literature review, this doesn't highlight any formal standards or frameworks (e.g. IFLA or, in the U.S. context ACRL). It seems to me what you're describing is information behavior of lecturers? That would require a different lit review

This study employed the relational perspective of information literacy which emphasis the experience of individuals with their information world. This experience would enable them "to draw upon in new situations", which can be translated as the way they learnt from that situation and applying them to a new situation. Bruce described the information literate person as being “one who experiences information literacy in a range of ways, and is able to determine the nature of experience it is necessary to draw upon in new situations” (1997a, p. 169). Therefore, information literacy is connected to this new situation experienced by the lecturers of this study. Moreover, the relational perspective of information literacy framework is adopted in this study and have been mentioned in the literature review section.

• I wasn't entirely clear on the research questions - I know you were investigating how the lecturers changed their teaching based on the transition to online education, but it wasn't clear to me how this fits into the overall research framework of either information literacy or online teaching. Again, for the latter, a different or supplemental lit review and framework would be needed

Thank you for point this up. The literature review section has been revised as suggested emphasizing the emergency remote teaching and information literacy framework.

• How do these particular lecturers represent the wider population? What were your questions? Methods need to be more specific.

This qualitative study revealed the perspective of lecturers in the Faculty of Humanities, Universitas Diponegoro regarding their online teaching experience. The findings of the study do not aim to generalize the whole population of lecturers. However, the Methods section has been revised as suggested by providing research question

---

## [Decision Letter · Decision Letter 1]

29 Jun 2021

PONE-D-20-39011R1

Lecturers' information literacy experience in remote teaching during COVID-19 pandemic

PLOS ONE

Dear Dr. Heriyanto,

Thank you for submitting your manuscript to PLOS ONE. After careful consideration, we feel that it has merit but does not fully meet PLOS ONE’s publication criteria as it currently stands. Therefore, we invite you to submit a revised version of the manuscript that addresses the points raised during the review process.

As suggested by the second reviewer, still more work is expected in reporting the analysis process of the qualitative data.

We look forward to receiving your revised manuscript.

Kind regards,

Mingming Zhou, Ph.D.

Academic Editor

PLOS ONE

Reviewers' comments:

Reviewer's Responses to Questions

**Comments to the Author**

1. If the authors have adequately addressed your comments raised in a previous round of review and you feel that this manuscript is now acceptable for publication, you may indicate that here to bypass the “Comments to the Author” section, enter your conflict of interest statement in the “Confidential to Editor” section, and submit your "Accept" recommendation.

Reviewer #1: All comments have been addressed

Reviewer #3: (No Response)

2. Is the manuscript technically sound, and do the data support the conclusions?

Reviewer #1: Partly

Reviewer #3: Partly

3. Has the statistical analysis been performed appropriately and rigorously? 

Reviewer #1: N/A

Reviewer #3: N/A

4. Have the authors made all data underlying the findings in their manuscript fully available?

Reviewer #1: Yes

Reviewer #3: No

5. Is the manuscript presented in an intelligible fashion and written in standard English?

Reviewer #1: Yes

Reviewer #3: Yes

6. Review Comments to the Author

Reviewer #1: I appreciate the changes to this manuscript. My concerns were addressed and the research question more clearly answered. While the data support the conclusion, there were instances where only one participant's concerns were addressed. It would have been helpful to see the questions from semi-structured interviews to get a better perspective.

Overall, however, I believe this manuscript is suitable for publication. Thank you for your extensive revisions!

Reviewer #3: Thank you for the opportunity to review this submission, and for the work you have done on this project. The manuscript reviewed focused on interview research meant to generate findings about the "information literacy experiences" of lecturers during the COVID-19 Pandemic. While I found several aspects of the manuscript interesting, including the introduction of research in the area of online teaching and learning that I was not quite as familiar with, there are several issues, including those raised by previous reviewers, that must be addressed before this manuscript is ready for publication.

Firstly, as reviewer 2 remarked, a much stronger conceptualization of information literacy must not only be developing in the introductory pages of the manuscript, but it must also be shown to have clearly guided the data analytic procedure. A good conceptual definition, at the very least, should provide a guideline for what DOES count as information literacy, as well as a clear demarcation of what is NOT information literacy. The relational framing provided, while interesting, is too broad to provide inclusion and exclusion criteria that will help focus and develop rigor in the data analysis.

On a related note, the discussion of "qualitative research" in the methodology is also quite broad; in some cases, this broadness is a mischaracterization of the diversity of methodological thought within research traditions who have been historically grouped together as "qualitative" (e.g. discourse analysis, grounded theory, phenomenology, narrative analysis, etc.) but are, in reality, founded in very different epistemological and ontological assumptions. My advice would be to clarify the particulars of the qualitative approach that you are taking (the thematic approach you refer to seems to draw on certain traditions of grounded theory, or at the least, constant-comparative analysis), but you should make these distinctions more clear in your discussion.

Finally, related to the comments of other reviewers, one challenge in presenting thematically analyzed research is making transparent the rigor and systematic nature of your analysis. If this is not clear to readers (or reviewers), then the findings of your work are called into question. My suggestion here would be to provide more clear examples of how you developed codes from your open readings of the data (stage 1), how those codes were revised, refined, and reapplied to the data (stage 2), and how those codes where them "thematized" to provide your findings (stage 3). It would help, in this case, to make some of your own assumptions and prior experiences (what some refer to as positionality) as interpretive researchers transparent so as to increase readers' ability to evaluate your claims and their subjectivities. In these cases, drawing on passive constructions of language in the hope of maintaining "objectivity" actually serves to obscure, rather than clarify your analysis.

I do think the study COULD warrant publication in this journal, but not until the above issues, as well as some of the concerns of other reviewers are more rigorously addressed. Thank you again for the opportunity to review this submission.

7. PLOS authors have the option to publish the peer review history of their article (what does this mean?). If published, this will include your full peer review and any attached files.

Reviewer #1: No

Reviewer #3: No

---

## [Author Response · Author response to Decision Letter 1]

1 Jul 2021

Reviewer 1:

I appreciate the changes to this manuscript. My concerns were addressed and the research question more clearly answered. While the data support the conclusion, there were instances where only one participant's concerns were addressed. It would have been helpful to see the questions from semi-structured interviews to get a better perspective.

Overall, however, I believe this manuscript is suitable for publication. Thank you for your extensive revisions!

Author comments

Thank you for the suggestion. We have followed this suggestion by providing the questions from semi-structured interviews in the method section.

Reviewer 3:

Firstly, as reviewer 2 remarked, a much stronger conceptualization of information literacy must not only be developing in the introductory pages of the manuscript, but it must also be shown to have clearly guided the data analytic procedure. A good conceptual definition, at the very least, should provide a guideline for what DOES count as information literacy, as well as a clear demarcation of what is NOT information literacy. The relational framing provided, while interesting, is too broad to provide inclusion and exclusion criteria that will help focus and develop rigour in the data analysis

Author comment:

Thank you for the suggestion. A more specific conceptual definition of relational perspective information literacy has been added to the literature review section as recommended. 

Reviewer 3:

On a related note, the discussion of "qualitative research" in the methodology is also quite broad; in some cases, this broadness is a mischaracterization of the diversity of methodological thought within research traditions who have been historically grouped together as "qualitative" (e.g. discourse analysis, grounded theory, phenomenology, narrative analysis, etc.) but are, in reality, founded in very different epistemological and ontological assumptions. My advice would be to clarify the particulars of the qualitative approach that you are taking (the thematic approach you refer to seems to draw on certain traditions of grounded theory, or at the least, constant-comparative analysis), but you should make these distinctions more clear in your discussion

Author comment:

Agree. This study employed basic qualitative approach following the recommendation by Merriam (2009). As the characteristic of basic qualitative is similar with all of qualitative research, other types of qualitative studies have additional dimension. For example, a grounded theory study seeks not just to understand, but also to build a substantive theory about the phenomenon being studied.

Reviewer 3:

Finally, related to the comments of other reviewers, one challenge in presenting thematically analyzed research is making transparent the rigor and systematic nature of your analysis. If this is not clear to readers (or reviewers), then the findings of your work are called into question. My suggestion here would be to provide more clear examples of how you developed codes from your open readings of the data (stage 1), how those codes were revised, refined, and reapplied to the data (stage 2), and how those codes where them "thematized" to provide your findings (stage 3). It would help, in this case, to make some of your own assumptions and prior experiences (what some refer to as positionality) as interpretive researchers transparent so as to increase readers' ability to evaluate your claims and their subjectivities. In these cases, drawing on passive constructions of language in the hope of maintaining "objectivity" actually serves to obscure, rather than clarify your analysis

Author comment:

Thank you for the valuable suggestion. A detailed description of thematic analysis process has been added to the method section for making transparent the rigor and systematic nature of the data analysis.

---

## [Decision Letter · Decision Letter 2]

10 Aug 2021

PONE-D-20-39011R2

Lecturers' information literacy experience in remote teaching during COVID-19 pandemic

PLOS ONE

Dear Dr. Heriyanto,

Thank you for submitting your manuscript to PLOS ONE. After careful consideration, we feel that it has merit but does not fully meet PLOS ONE’s publication criteria as it currently stands. Therefore, we invite you to submit a revised version of the manuscript that addresses the points raised during the review process.

While one reviewer is happy with the revision, the other reviewer proposed further questions for the authors to think and revise. Further, having the manuscript proofread by a language editor would also help.

We look forward to receiving your revised manuscript.

Kind regards,

Mingming Zhou, Ph.D.

Academic Editor

PLOS ONE

Journal Requirements:

Reviewers' comments:

Reviewer's Responses to Questions

**Comments to the Author**

1. If the authors have adequately addressed your comments raised in a previous round of review and you feel that this manuscript is now acceptable for publication, you may indicate that here to bypass the “Comments to the Author” section, enter your conflict of interest statement in the “Confidential to Editor” section, and submit your "Accept" recommendation.

Reviewer #1: (No Response)

Reviewer #3: All comments have been addressed

2. Is the manuscript technically sound, and do the data support the conclusions?

Reviewer #1: No

Reviewer #3: Yes

3. Has the statistical analysis been performed appropriately and rigorously? 

Reviewer #1: N/A

Reviewer #3: Yes

4. Have the authors made all data underlying the findings in their manuscript fully available?

Reviewer #1: No

Reviewer #3: Yes

5. Is the manuscript presented in an intelligible fashion and written in standard English?

Reviewer #1: No

Reviewer #3: Yes

6. Review Comments to the Author

Reviewer #1: This is an interesting and timely study, particularly since the conditions were unique.

I do believe, however, that the findings are rather vague and confusing. Below are some issues that I noted throughout the paper:

The first sentence of the introduction needs a citation.

In the literature review, it says that the current study investigated the subject matter, rather than issues. Then later it says they are investigating the lecturers’ information literacy experience. This doesn’t make sense to me.

The authors frequently refer to the perspectives of the “majority of lecturers.” Why not give exact numbers since there were only 15 participants?

I’m confused by your statement that lecturers invited students to text discussions after video meetings. Do you mean text questions? Then you stated that students exhibited various behaviors during the online session. Can you be more specific? What behaviors do you mean? If you are referring to whether or not they participated in the chat or asked questions, you should specify that.

While you used thematic analysis for your methodology, I’m not seeing any clear explanation of themes. You describe the perspectives of individual or “majority” of lecturers, but do not connect those to any type of theme.

I do not see the results to be significant since they seem to align with past research.

Last, there are many grammatical errors throughout, possibly because of translations. However, these errors make the manuscript difficult to follow.

Reviewer #3: I appreciate the careful work the authors have done to address my concerns. I look forward to seeing this in print and discussing with my graduate students.

7. PLOS authors have the option to publish the peer review history of their article (what does this mean?). If published, this will include your full peer review and any attached files.

Reviewer #1: No

Reviewer #3: **Yes: **Earl Aguilera

---

## [Author Response · Author response to Decision Letter 2]

21 Aug 2021

The reference list has been reviewed. As suggested, I have not cited the retracted paper. However, I have removed the preprint paper and replaced it with a relevant paper. The new paper listed in the reference list is 

• Pal D, Vanijja V. Perceived usability evaluation of Microsoft Teams as an online learning platform during COVID-19 using system usability scale and technology acceptance model in India. Child Youth Serv Rev [Internet]. 2020;119(October):105535. Available from: https://doi.org/10.1016/j.childyouth.2020.105535

1. Thank you for raising this point up. The sentence has been revised and a relevant citation has been added.

2. Yes, I agree with the reviewer that the statements are a bit confusing. The first statement has been revised in order to cohere with the second statement that the study is about investigating lecturers’ information literacy experience. 

3. The word ‘majority’ has been replaced with the exact name or number of the participants

4. I read the sentence again and I agree with you that this sentence should be more specific. More specific statements have been provided to the paragraph. It explains the behavior of students during the online conversation and text discussions, where some students contribute actively during the video meeting but some prefer to be active in the text discussion. 

5. A description of each theme has been provided to the results section. Further, the significance of the results has also been provided to the discussion section.

6. The revised version of the manuscript has been proofread by professional proofreading services to improve the clarity and readability.

---

## [Decision Letter · Decision Letter 3]

2 Nov 2021

Lecturers' information literacy experience in remote teaching during the COVID-19 pandemic

PONE-D-20-39011R3

Dear Dr. Heriyanto,

We’re pleased to inform you that your manuscript has been judged scientifically suitable for publication and will be formally accepted for publication once it meets all outstanding technical requirements.

Kind regards,

Mingming Zhou, Ph.D.

Academic Editor

PLOS ONE

Additional Editor Comments (optional):

Reviewers' comments:

Reviewer's Responses to Questions

**Comments to the Author**

1. If the authors have adequately addressed your comments raised in a previous round of review and you feel that this manuscript is now acceptable for publication, you may indicate that here to bypass the “Comments to the Author” section, enter your conflict of interest statement in the “Confidential to Editor” section, and submit your "Accept" recommendation.

Reviewer #3: All comments have been addressed

2. Is the manuscript technically sound, and do the data support the conclusions?

Reviewer #3: Yes

3. Has the statistical analysis been performed appropriately and rigorously? 

Reviewer #3: N/A

4. Have the authors made all data underlying the findings in their manuscript fully available?

Reviewer #3: Yes

5. Is the manuscript presented in an intelligible fashion and written in standard English?

Reviewer #3: Yes

6. Review Comments to the Author

Reviewer #3: I appreciate the changes to this manuscript. My concerns were addressed and the research question more clearly answered.

Overall, I believe this manuscript is suitable for publication. Thank you for your extensive revisions.

7. PLOS authors have the option to publish the peer review history of their article (what does this mean?). If published, this will include your full peer review and any attached files.

Reviewer #3: No

---

## [Editor Report · Acceptance letter]

10 Mar 2022

PONE-D-20-39011R3 

Lecturers’ information literacy experience in remote teaching during the COVID-19 pandemic 

Dear Dr. -:

I'm pleased to inform you that your manuscript has been deemed suitable for publication in PLOS ONE. Congratulations! Your manuscript is now with our production department. 

Kind regards, 

on behalf of

Dr. Mingming Zhou 

Section Editor

PLOS ONE